# Emerging Strategies in Cartilage Repair and Joint Preservation

**DOI:** 10.3390/medicina61010024

**Published:** 2024-12-27

**Authors:** Mircea Adrian Focsa, Sorin Florescu, Armand Gogulescu

**Affiliations:** Faculty of Medicine, Victor Babes University of Medicine and Pharmacy, 2 Eftimie Murgu, 300041 Timisoara, Romania; mfocsa@umft.ro (M.A.F.); gogulescu.armand@umft.ro (A.G.)

**Keywords:** cartilage repair, regenerative medicine, mesenchymal stem cells, gene therapy, biomaterials, tissue engineering, joint preservation

## Abstract

*Background and Objectives:* Cartilage repair remains a critical challenge in orthopaedic medicine due to the tissue’s limited self-healing ability, contributing to degenerative joint conditions such as osteoarthritis (OA). In response, regenerative medicine has developed advanced therapeutic strategies, including cell-based therapies, gene editing, and bioengineered scaffolds, to promote cartilage regeneration and restore joint function. This narrative review aims to explore the latest developments in cartilage repair techniques, focusing on mesenchymal stem cell (MSC) therapy, gene-based interventions, and biomaterial innovations. It also discusses the impact of patient-specific factors, such as age, defect size, and cost efficiency, on treatment selection and outcomes. *Materials and Methods:* This review synthesises findings from recent clinical and preclinical studies published within the last five years, retrieved from the PubMed, Scopus, and Web of Science databases. The search targeted key terms such as “cartilage repair”, “stem cell therapy”, “gene editing”, “biomaterials”, and “tissue engineering”. *Results:* Advances in MSC-based therapies, including autologous chondrocyte implantation (ACI) and platelet-rich plasma (PRP), have demonstrated promising regenerative potential. Gene-editing tools like CRISPR/Cas9 have facilitated targeted cellular modifications, while novel biomaterials such as hydrogels, biodegradable scaffolds, and 3D-printed constructs have improved mechanical support and tissue integration. Additionally, biophysical stimuli like low-intensity pulsed ultrasound (LIPUS) and electromagnetic fields (EMFs) have enhanced chondrogenic differentiation and matrix production. Treatment decisions are influenced by patient age, cartilage defect size, and financial considerations, highlighting the need for personalised and multimodal approaches. *Conclusions:* Combining regenerative techniques, including cell-based therapies, gene modifications, and advanced scaffolding, offers a promising pathway towards durable cartilage repair and joint preservation. Future research should focus on refining integrated therapeutic protocols, conducting long-term clinical evaluations, and embracing personalised treatment models driven by artificial intelligence and predictive algorithms.

## 1. Introduction

Cartilage is a specialised connective tissue essential for joint function, providing a smooth, lubricated surface for articulation and effective load distribution. Its avascular nature and limited intrinsic repair capacity make it particularly susceptible to damage from injuries or degenerative conditions, often leading to progressive joint dysfunction and osteoarthritis (OA), which significantly contribute to global disability.

The morphopathologic mechanisms underlying knee cartilage damage involve a complex interplay of mechanical, biochemical, and cellular factors leading to the degradation of the extracellular matrix (ECM). Mechanical stress from acute injuries or repetitive overuse can disrupt the collagen network and deplete proteoglycan content, compromising the structural integrity of cartilage [1]. Concurrently, inflammatory mediators such as cytokines and matrix metalloproteinases (MMPs) are upregulated, accelerating ECM breakdown and inhibiting repair processes [2]. Chondrocyte apoptosis further diminishes the tissue’s capacity for maintenance and regeneration, culminating in progressive cartilage deterioration and joint dysfunction [3].

Recent epidemiological data indicate a concerning rise in knee cartilage injuries among younger individuals, particularly athletes. This trend is attributed to increased participation in high-intensity sports, leading to a higher incidence of acute traumatic events like anterior cruciate ligament (ACL) tears and meniscal injuries, which are often associated with concomitant cartilage damage [4]. Moreover, repetitive microtrauma from overuse without adequate recovery can initiate degenerative changes in cartilage [5]. The growing popularity of high-intensity sports and year-round athletic participation has led to more frequent exposure to mechanical stress on the joints. Inadequate recovery periods, improper training techniques, and limited access to preventive sports medicine programmes further exacerbate the risk of injuries. Additionally, advancements in diagnostic imaging technologies have improved the detection of cartilage damage, resulting in a higher reported incidence. The growing prevalence of obesity in youth populations also contributes to elevated joint loading, exacerbating cartilage wear [6]. These factors underscore the need for targeted preventive strategies and early interventions to address cartilage damage in younger demographics.

Traditional treatments often focus on symptom management, whereas regenerative approaches aim to restore the damaged cartilage to its original state. Recent advancements in regenerative medicine have introduced innovative therapies aimed at restoring cartilage structure and function. Platelet-rich plasma (PRP) and mesenchymal stem cell (MSC) therapies have emerged as pivotal nonoperative strategies. PRP leverages growth factors to modulate inflammation, stimulate angiogenesis, and enhance tissue healing, while MSCs offer multipotent capabilities, immunomodulation, and paracrine effects to promote cartilage regeneration. These approaches have shown promise in preclinical and early clinical studies, demonstrating improvements in pain relief, joint function, and cartilage quality [7,8,9].

However, the clinical translation of PRP and MSC therapies faces significant challenges. The heterogeneity in PRP preparation protocols and MSC sourcing, alongside variability in therapeutic outcomes, underscores the need for standardisation and optimisation [10,11]. Moreover, PRP’s efficacy is often limited by its short-lived bioactivity, while MSC therapies face regulatory hurdles, scalability issues, and concerns about the long-term safety of cell-based interventions [12,13].

Despite these advancements, unmet needs persist in achieving durable and reproducible cartilage repair. Current therapies often fail to restore hyaline cartilage, the native cartilage type, instead producing fibrocartilage with inferior mechanical and functional properties. Furthermore, many treatments are stage specific and less effective in advanced cartilage damage or osteoarthritis, leaving a critical gap for patients with severe joint degeneration [14,15].

These challenges highlight the necessity for next-generation regenerative techniques that address the limitations of existing therapies. Novel approaches such as exosome-based cell-free therapies, hybrid biomaterial scaffolds, and induced pluripotent stem cell (iPSC)-derived cartilage constructs hold immense potential to overcome these barriers [16]. These strategies aim to redefine the landscape of cartilage repair and joint preservation by integrating advances in bioengineering, precision medicine, and computational modelling.

This narrative review explores the transformative potential of these emerging regenerative techniques, emphasising their ability to address the unmet needs in cartilage repair. We provide an updated overview of advancements in orthobiologics and biomaterials, focusing on their clinical relevance, challenges, and the path forward towards achieving durable and reproducible cartilage regeneration.

This review uniquely integrates cutting-edge regenerative strategies, including gene editing, bioengineered scaffolds, and personalised cell-based therapies, emphasising both technical innovations and clinical applications. By considering patient-specific factors such as age, defect size, and healthcare accessibility, our review offers a practical perspective not commonly addressed in similar publications.

## 2. Article Selection Process

We conducted a comprehensive literature review by searching scientific databases, including PubMed, Scopus, and Web of Science. The search focused on peer-reviewed articles published in the last five years (2019–2024) to ensure the inclusion of the latest advancements in cartilage tissue engineering and regenerative medicine. Search terms included “cartilage repair”, “stem cell therapy”, “gene editing”, “biomaterials”, and “tissue engineering”.

We included articles that reported experimental, clinical, or preclinical studies relevant to cartilage repair techniques, focusing on innovative and emerging strategies. Systematic reviews, narrative reviews, and meta-analyses were also considered when they provided comprehensive overviews of the relevant technologies. Studies published in non-English languages, conference abstracts, and those lacking significant experimental data were excluded.

Relevant data, including therapeutic approaches, experimental outcomes, clinical applicability, and limitations, were extracted from selected articles. Key findings were synthesised into thematic sections covering the key cartilage regenerative techniques. Potential challenges and future research directions were critically analysed.

## 3. Key Regenerative Techniques in Cartilage Repair

Recent advancements in regenerative medicine have led to the emergence of groundbreaking techniques aimed at restoring damaged cartilage by enhancing the body’s natural healing processes. These innovative methods, including stem cell therapy and tissue engineering, not only focus on repairing cartilage but also promote long-term joint health. By leveraging the body’s inherent capabilities, we can offer effective and less invasive solutions for individuals suffering from cartilage injuries and degenerative conditions. Embracing these advancements holds the promise of improved mobility and a better quality of life, highlighting the importance of the further exploration of these promising treatments (Figure 1).

The **microfracture technique** is a widely utilised surgical method for repairing articular cartilage defects, particularly in the knee. It involves creating small perforations, or microfractures, in the subchondral bone beneath the damaged cartilage. This process allows blood and bone marrow elements, including mesenchymal stem cells (MSCs), to migrate into the defect site, forming a fibrin clot that serves as a scaffold for new tissue growth. However, the reparative tissue generated is often fibrocartilage, which lacks the durability and mechanical properties of native hyaline cartilage, leading to concerns about the longevity and effectiveness of the repair. To address these limitations, several novel approaches have been developed to augment the microfracture technique [17] by promoting the formation of more durable, hyaline-like cartilage, thereby enhancing the longevity and functionality of the repair.

**Autologous Matrix-Induced Chondrogenesis (AMIC):** This one-step procedure combines microfracture with the application of a collagen I/III membrane over the defect. The membrane stabilises the blood clot and enhances the differentiation of MSCs into chondrocytes, promoting the formation of hyaline-like cartilage. AMIC is less complex and more cost effective than cell-based therapies, as it does not require cell harvesting or laboratory expansion. Clinical studies have reported improved outcomes with AMIC compared to microfracture alone, particularly in terms of defect filling and tissue quality [18].

**Scaffold-Augmented Microfracture:** In this approach, scaffolds are placed over the microfracture area to provide structural support and a conducive environment for cell proliferation and differentiation. Innovations include 3D bioprinting and hydrogels that mimic the natural cartilage environment, facilitating tissue regeneration. Scaffolds can be composed of various materials, including collagen, hyaluronic acid, or synthetic polymers, and may be combined with bioactive factors to enhance cartilage regeneration further. Studies have shown that scaffold augmentation can lead to superior repair tissue quality and improved clinical outcomes compared to microfracture alone [19]. Recent advancements have focused on three primary types of scaffolds: hydrogels, biodegradable polymers, and natural polymers.

*Hydrogels* offer a three-dimensional, hydrated network that closely mimics the extracellular matrix (ECM) of cartilage, facilitating cell attachment, proliferation, and differentiation. Their high water content and tunable mechanical properties make them suitable for encapsulating cells and delivering bioactive molecules, thereby promoting tissue regeneration. Recent developments in hydrogel technology have enhanced their mechanical strength and biocompatibility, making them more effective in cartilage repair applications [20].

*Biodegradable polymers*, such as polylactic acid (PLA) and polyglycolic acid (PGA), provide temporary structural support to the regenerating tissue and degrade over time, reducing the need for surgical removal (Figure 2). Their degradation rates can be tailored to match the tissue regeneration process, ensuring seamless integration with native tissue. Advances in polymer chemistry have led to the development of copolymers and composites that exhibit improved mechanical properties and controlled degradation rates, enhancing their applicability in cartilage tissue engineering [21].

*Natural polymers*, including collagen and hyaluronic acid, are inherently biocompatible and bioactive, closely resembling the native ECM. Scaffolds derived from these materials support cell adhesion, proliferation, and differentiation, facilitating the formation of functional cartilage tissue. Innovations in processing techniques have enabled the fabrication of natural polymer-based scaffolds with enhanced mechanical strength and structural integrity, making them more viable for clinical applications [22].

Recent advancements in 4D bioprinting are creating new opportunities for cartilage regeneration by enabling the production of dynamic, stimuli-responsive scaffolds that can adapt to their physiological environment over time. These scaffolds can undergo controlled shape transformations, altering their mechanical and biological properties in response to external stimuli such as temperature, pH, or mechanical forces [23]. Researchers also developed magnetic constructs using 4D bioprinting with silk fibroin in gelatine bioinks [24]. These shape-morphing constructs can change their shape when exposed to an external magnetic field, which enhances their integration into irregular cartilage defects. This adaptive behaviour promotes better integration, long-term stability, and functionality in cartilage repair applications, positioning 4D bioprinting as a transformative tool in regenerative medicine. 

**Biological Augmentation with Bone Marrow Aspirate Concentrate (BMAC):** This technique involves applying BMAC, which is rich in MSCs and growth factors, to the microfracture defect. The combination aims to enhance the regenerative potential by increasing the concentration of progenitor cells at the repair site. Preliminary studies suggest that BMAC augmentation can improve the quality of the repair tissue and clinical outcomes, though further research is needed to establish its efficacy [25].

**Autologous Chondrocyte Implantation (ACI):** ACI involves harvesting chondrocytes from a non-weight-bearing area of the patient’s joint, expanding them in vitro, and then implanting them into the cartilage defect. This technique aims to regenerate hyaline-like cartilage, offering improved outcomes over fibrocartilage repair. However, it requires two surgical procedures and presents challenges related to cell dedifferentiation during expansion [26].

**Matrix-Induced Autologous Chondrocyte Implantation (MACI):** An evolution of ACI, MACI is a two-stage procedure. Initially, healthy cartilage is harvested arthroscopically from a non-weight-bearing area of the patient’s joint. The chondrocytes are then isolated and expanded in vitro. These cultured chondrocytes are seeded onto a biodegradable collagen scaffold, which is later implanted into the cartilage defect during a second surgical procedure. The scaffold facilitates the integration and maturation of the implanted chondrocytes, aiming to regenerate hyaline-like cartilage. MACI allows for the implantation of a higher number of chondrocytes directly into the defect, potentially leading to better-quality cartilage repair. It is particularly beneficial for larger defects that may not respond adequately to microfracture [27].

**Osteochondral Autograft Transplantation (OAT):** This technique transplants cylindrical plugs of healthy cartilage and underlying bone from a non-essential area of the joint to the damaged site. OAT is beneficial for small defects and provides immediate structural support with hyaline cartilage. However, donor site morbidity and limited availability of graft material are concerns.

**Osteochondral Allograft Transplantation:** Similar to OAT, this method uses donor tissue from cadaveric sources to repair larger cartilage defects. It allows for the transplantation of mature hyaline cartilage but carries risks of immune rejection and disease transmission.

**Platelet-Rich Plasma (PRP) Therapy:** PRP involves concentrating platelets from the patient’s own blood and injecting them into the affected joint. Platelets release growth factors that modulate inflammation, stimulate angiogenesis, and promote tissue healing. Clinical studies have shown that PRP can improve pain and function in patients with cartilage injuries and early osteoarthritis [28]. However, outcomes may vary due to differences in PRP preparation methods and individual patient factors, with the best results being obtained by leukocyte-poor PRP. The double-spin protocol, which involves processing 10 mL of blood at 100× *g* and then at 1600× *g* for 20 min each in a 15 mL tube and using the lower one-third of the final product, has shown consistently high platelet recovery rates (86–99%) and a concentration of 6×. This method also preserves both the integrity and viability of the platelets without disruption [29]. In a study conducted by Bansal et al. (2021) [11], it was confirmed that 10 billion platelets have long-term effects on moderate knee osteoarthritis.

**Mesenchymal Stem Cell (MSC) Therapy:** MSCs are multipotent cells capable of differentiating into various cell types, including chondrocytes, the primary cells in cartilage. When injected into a damaged joint, MSCs can potentially regenerate cartilage through differentiation and by secreting bioactive molecules that modulate the local environment. Clinical evidence suggests that MSC therapy can reduce pain and improve joint function, but challenges remain regarding cell sourcing, delivery methods, and ensuring consistent outcomes [30]. The goal of next-generation therapy that uses mesenchymal stem cells (MSCs) is not simply to optimise treatment based on variations in tissue or cell quantity. Instead, it aims to identify the stage of cartilage damage at which these specialised cells are essential for promoting cartilage regeneration.

MSC therapy utilises cells from various tissues, each offering distinct advantages. Bone marrow-derived MSCs (BM-MSCs), traditionally used in regenerative medicine, are obtained from bone marrow aspirates. However, their limited ability to proliferate and the invasive nature of extraction present challenges. In contrast, adipose-derived MSCs (AD-MSCs) provide a higher yield and are easier to obtain than BM-MSCs. Their potential for chondrogenesis makes them a suitable alternative for cartilage repair. Umbilical cord-derived MSCs (UC-MSCs), which are sourced from Wharton’s jelly of the umbilical cord, demonstrate strong proliferation and differentiation capabilities. Notably, they can undergo extensive passages while maintaining their multipotency, making them a promising cell source for cartilage regeneration [31].

Enhancing mesenchymal stem cell (MSC)-based therapies for cartilage repair involves optimising various factors to improve chondrogenic differentiation and tissue regeneration. Key strategies include the following:

**Growth Factors and Cytokines:** Recent studies have underscored the pivotal roles of growth factors and cytokines in cartilage repair. Transforming growth factor-beta (TGF-β) is essential for initiating chondrogenesis by promoting mesenchymal stem cell differentiation into chondrocytes and maintaining their phenotype. Bone morphogenetic proteins (BMPs), particularly BMP-2 and BMP-7, enhance cartilage matrix production and chondrocyte maturation, contributing to cartilage homeostasis and repair [32]. Insulin-like growth factor-1 (IGF-1) stimulates the synthesis of proteoglycans and collagen, supporting the formation and maintenance of the cartilage matrix [33]. Collectively, these factors orchestrate the complex processes of cartilage development and regeneration.

A study by Lin et al. [34] demonstrated that ghrelin significantly increased the expression of chondrogenic markers such as SOX9, ACAN, and COL II in MSCs. When combined with TGF-β, ghrelin synergistically activated the phosphorylation of ERK1/2 and DNMT3A, further enhancing chondrogenic gene expression. In vivo experiments using a rat osteochondral defect model showed that delivering ghrelin alongside TGF-β significantly improved cartilage regeneration compared to TGF-β alone. Another study developed a microsphere/hydrogel system for the dual delivery of TGF-β3 and ghrelin. This system provided sustained release of both growth factors, promoting cartilage tissue formation and significantly enhancing the chondrogenic differentiation of MSCs. These findings suggest that the combined application of ghrelin and TGF-β can improve the efficacy of MSC-based therapies for cartilage repair by enhancing chondrogenic differentiation and tissue regeneration.

However, therapeutic application of the growth factors is limited by short half-lives, uncontrolled release, and potential off-target effects. Future research should focus on developing sustained-release systems such as encapsulated hydrogels, gene-activated matrices, and tissue-specific delivery platforms to enhance the localised and long-term bioactivity of these signalling molecules.

**Mechanical stimulation** plays a crucial role in cartilage tissue engineering by enhancing extracellular matrix synthesis and promoting chondrogenic differentiation. Dynamic compression, which involves applying mechanical loads to engineered tissue constructs, has been shown to improve the mechanical properties of the tissue and to stimulate the production of cartilage-specific matrix components. Similarly, shear stress, generated through fluid flow or direct mechanical forces, encourages chondrogenic differentiation and increases matrix production, thereby contributing to the development of functional cartilage tissue. These mechanical cues are essential for replicating the native cartilage environment [35,36]. However, mechanical stimulation lacks standardisation regarding optimal intensity, frequency, and duration for different patient profiles. Future studies should define precise stimulation protocols and explore combined physical and biochemical stimulation approaches.

**Biophysical stimuli** have been shown to enhance cartilage regeneration significantly. Low-intensity pulsed ultrasound (LIPUS) is a non-invasive therapy that promotes chondrogenic differentiation and cartilage matrix production by inhibiting the TNF signalling pathway, thereby facilitating articular cartilage regeneration. This modality offers a non-invasive adjunct to other regenerative treatments [37]. While there is no universal standard governing all aspects of LIPUS application, certain parameters have been widely adopted in clinical and research settings: a frequency of 1.5 MHz, a pulse duration of approximately 200 microseconds, a pulse repetition frequency of around 1 kHz, and a spatial average temporal average (SATA) intensity of 30 mW/cm^2^. These settings are exemplified by the Exogen device, which operates at these specifications and has received approval from the U.S. Food and Drug Administration (FDA) for bone healing applications [38].

Similarly, **electromagnetic fields (EMFs)** regulate calcium-mediated cell fate decisions in stem cells, enhancing chondrogenesis and contributing to effective cartilage repair [39]. Additionally, pulsed electromagnetic fields (PEMFs) have been found to potentiate the chondrogenesis of bone marrow mesenchymal stem cells by regulating the Wnt/β-catenin signalling pathway, thereby contributing to effective cartilage repair [40].

**Genetic modification** techniques have significantly advanced cartilage tissue engineering by enhancing the chondrogenic potential of mesenchymal stem cells (MSCs). The overexpression of chondrogenic genes, such as SOX9, directs MSCs towards a chondrogenic lineage, promoting cartilage formation. For instance, SOX9 overexpression has been shown to potentiate BMP2-induced chondrogenic differentiation while inhibiting osteogenic differentiation, thereby facilitating effective cartilage regeneration [41]. Additionally, gene editing tools like CRISPR/Cas9 enable precise genomic modifications to augment regenerative capabilities. Recent studies have demonstrated that CRISPR/Cas9-mediated knockout of specific genes, such as RUNX2, in human MSCs can lead to the production of extracellular matrices with superior quality, enhancing cartilage repair outcomes [42].

**Co-culture systems**, which involve cultivating mesenchymal stem cells (MSCs) alongside chondrocytes or other supportive cell types, have been shown to enhance chondrogenic differentiation and extracellular matrix production through paracrine signalling. This interaction facilitates the exchange of soluble factors and direct cell–cell contact, promoting a microenvironment conducive to cartilage regeneration. Studies have demonstrated that co-culturing MSCs with chondrocytes can lead to improved cartilage tissue formation compared to monocultures, highlighting the potential of co-culture strategies in advancing cartilage tissue engineering [43].

**Hypoxic Conditions:** Mimicking the low-oxygen environment of cartilage tissue can promote chondrogenic differentiation and matrix synthesis. Under hypoxia, MSCs exhibit increased expression of chondrogenic markers such as SOX9, aggrecan, and collagen type II, which are essential for cartilage formation. Additionally, hypoxia-inducible factors (HIFs), particularly HIF-1α, are stabilised in low-oxygen conditions, leading to the upregulation of genes that support chondrogenesis and ECM production. This hypoxia-driven pathway not only enhances the deposition of cartilage-specific ECM components but also improves the mechanical properties of engineered cartilage tissues, making hypoxic preconditioning a valuable strategy in regenerative medicine [44]. Maintaining controlled hypoxic conditions in clinical applications poses significant challenges due to the variability in tissue oxygen levels. Although hypoxic conditions stimulate chondrogenic differentiation and extracellular matrix production, prolonged hypoxia can induce tissue necrosis due to oxygen deprivation. Therefore, developing precise oxygen-regulating systems such as dynamic bioreactors and oxygen-sensitive biomaterials remains essential for engineering biomimetic conditions.

**Nutritional Supplements:** Ascorbic acid (vitamin C) is essential for collagen synthesis, serving as a cofactor for prolyl and lysyl hydroxylases, enzymes critical for stabilising the collagen triple-helix structure. A deficiency in vitamin C can impair collagen synthesis, compromising cartilage integrity. Additionally, glucosamine and chondroitin sulphate, key components of glycosaminoglycans in hyaline cartilage, have been shown to support cartilage matrix production. Glucosamine promotes the chondrogenic phenotype in both chondrocytes and mesenchymal stem cells, enhancing the synthesis of cartilage-specific extracellular matrix components. These supplements are commonly used as chondroprotective agents to maintain cartilage health and potentially slow the progression of degenerative joint diseases. A randomised, double-blind, placebo-controlled clinical study by Suryawanshi et al. (2024) demonstrated clinical improvement in patients with knee OA supplemented with a nutritional formula containing botanical actives and micronutrients, supporting the efficacy of such interventions in joint health management [45]. However, the efficacy of nutritional supplements remains inconclusive due to inconsistent clinical data and varied patient responses. Future research should involve large-scale, randomised clinical trials to establish evidence-based supplementation protocols.

Implementing these strategies can significantly improve the outcomes of MSC-based therapies for cartilage repair by creating an optimal environment for cell differentiation and tissue regeneration. Despite promising results, challenges such as limited cell survival, immune responses, and variability in patient outcomes persist. Future research should explore cell preconditioning, genetic modification, and personalised cell sourcing to improve efficacy and durability.

**Exosome-Based Therapies:** Exosomes are small extracellular vesicles secreted by cells, including MSCs, that carry proteins, lipids, and nucleic acids. They play a role in cell-to-cell communication and can modulate immune responses and tissue regeneration. Recent research indicates that exosomes can mimic the therapeutic effects of MSCs, offering a cell-free alternative for cartilage repair. Advantages include a reduced risk of immune rejection and easier storage and handling compared to live cells [46]. However, challenges persist in standardising exosome isolation, scaling up production, and ensuring consistent therapeutic potency. Future research should focus on optimising isolation techniques, enhancing targeted delivery through engineered exosomes, and conducting large-scale clinical trials to validate safety and efficacy in cartilage repair.

**Gene therapy** offers a novel approach to treating osteoarthritis (OA) by delivering therapeutic nucleic acids to target cells within the joint, enabling the modulation of complex disease pathways such as inflammation, cartilage degradation, and tissue repair. Using viral and non-viral vectors, gene therapy strategies can enhance the expression of anti-inflammatory cytokines like IL-1Ra and IL-10 or suppress catabolic enzymes such as MMP-13 and ADAMTS-5. The technology also extends to modifying intracellular signalling pathways through gene editing tools like CRISPR/Cas9, promising targeted and sustained therapeutic effects. Although still largely experimental, clinical trials have demonstrated the potential of gene therapies, such as intraarticular injections of adeno-associated virus (AAV) or lentivirus (LV) vectors [47]. However, gene editing for cartilage repair faces concerns related to delivery efficiency, off-target effects, and regulatory approval. Advancements in non-viral vectors and CRISPR-based precision editing are essential for safe and effective clinical translation.

## 4. Discussion

Cartilage repair has witnessed remarkable advancements, reflecting a growing emphasis on regenerative medicine and personalised therapies. While effective for symptomatic relief, traditional treatments often fall short in addressing the underlying structural damage in articular cartilage. This review has highlighted key emerging strategies, including cell-based therapies, gene editing, and innovative biomaterials, which hold the potential to revolutionise cartilage repair and joint preservation.

This review has several limitations. As a narrative review, it does not include a systematic analysis or meta-analysis, which limits the ability to quantify treatment efficacy. Additionally, the heterogeneity of the included studies and the evolving nature of regenerative technologies may influence the generalisability of findings. Future work should explore systematic approaches and comparative clinical trials to establish more definitive conclusions.

The development of advanced cartilage repair strategies has ushered in a new era in musculoskeletal medicine, focusing on personalised, regenerative, and minimally invasive approaches. Despite considerable progress, the adoption of a specific technique—or a combination of methods—depends on several critical factors, including patient-specific considerations, clinical presentation, and healthcare cost efficiency.

**Patient Age and Biological Potential:** Patient age is a pivotal factor influencing the selection of cartilage repair techniques. Younger patients generally exhibit a greater regenerative capacity due to higher cellular activity and a more responsive immune system. Techniques such as autologous chondrocyte implantation (ACI) or matrix-induced autologous chondrocyte implantation (MACI) are often preferred in younger individuals due to their capacity for long-term cartilage regeneration. In contrast, older patients may benefit more from minimally invasive therapies such as mesenchymal stem cell (MSC) injections or osteochondral allograft transplantation, in which the intrinsic healing potential is more limited.

**Size and Stage of the Cartilage Defect:** The size and stage of the cartilage lesion are crucial in determining the therapeutic approach. Small-to-medium defects are typically managed with microfracture, osteochondral autograft transplantation (OAT), or scaffold-based approaches. Larger defects or lesions involving subchondral bone may require more complex procedures, including osteochondral allografts or tissue-engineered constructs. Early-stage defects are amenable to cell-based therapies and minimally invasive procedures, while advanced osteoarthritic lesions may necessitate joint resurfacing or even arthroplasty.

**Cost Efficiency and Resource Availability:** The economic aspect of cartilage repair cannot be overlooked. While highly effective, cell-based therapies such as ACI or MACI are resource-intensive due to laboratory cell expansion, prolonged recovery, and multiple surgeries. Comparatively, injectable therapies like PRP or MSC-based treatments are less costly and require fewer clinical resources, making them appealing for patients in resource-constrained settings. Combining techniques, such as microfracture with biological augmentation (e.g., platelet-rich plasma or BMAC), can balance cost with therapeutic efficacy.

**Long-Term Clinical Outcomes:** The choice of therapy also hinges on the expected long-term outcomes. For example, while microfracture is widely used, it often leads to fibrocartilage formation, which lacks the mechanical integrity of hyaline cartilage and may deteriorate over time. In contrast, scaffold-based therapies, gene-modified MSCs, and tissue-engineered implants show potential for more durable hyaline-like cartilage regeneration. Long-term clinical trials are critical in defining which strategies offer sustained benefits.

**Combined and Multimodal Approaches:** The emerging evidence supports the integration of multiple techniques to maximise therapeutic outcomes. For example, combining gene therapy with scaffold-based cell delivery systems can enhance tissue regeneration, while mechanical stimulation through low-intensity pulsed ultrasound (LIPUS) or electromagnetic fields (EMFs) can further stimulate cellular activity and matrix production. Hypoxic preconditioning of MSCs or co-culture systems involving chondrocytes can also augment the repair process.

These factors underscore the need for a personalised approach in cartilage repair, guided by patient-specific considerations, clinical staging, and available healthcare resources. As the field advances, standardised treatment algorithms incorporating predictive models, artificial intelligence-driven decision-making tools, and long-term clinical data will be essential for optimising patient outcomes and advancing cartilage repair protocols.

## 5. Conclusions

The evolving landscape of cartilage repair is characterised by continuous innovation in regenerative therapies, including stem cell-based approaches, gene editing, bioengineered scaffolds, and advanced stimulation techniques. While each method has shown potential individually, combining them into comprehensive, multimodal treatment strategies may provide more durable and effective clinical outcomes. The development of personalised therapies guided by advances in genomics, biomaterials science, and bioengineering could redefine cartilage repair and joint preservation.

Future research should focus on refining combined approaches, developing standardised treatment protocols, and conducting long-term clinical trials to evaluate safety, efficacy, and sustainability. Interdisciplinary collaboration between clinicians, bioengineers, and regulatory bodies will be essential to translate these advanced therapies into routine clinical practice. Personalised, data-driven models guided by artificial intelligence and predictive algorithms will likely shape the next generation of cartilage repair solutions, ultimately enhancing patient outcomes and redefining standards in joint preservation and musculoskeletal care.

## Figures and Tables

**Figure 1 medicina-61-00024-f001:**
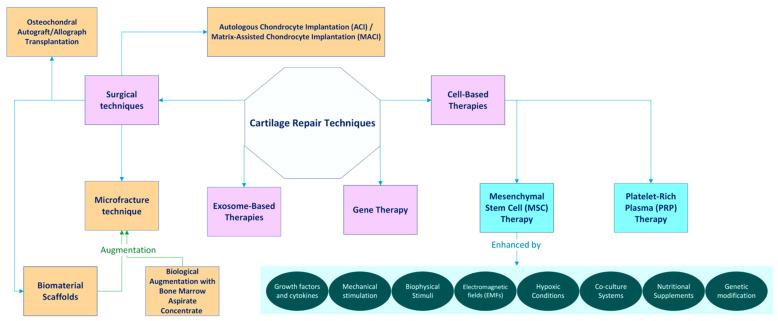
Overview of key regenerative techniques in cartilage repair.

**Figure 2 medicina-61-00024-f002:**
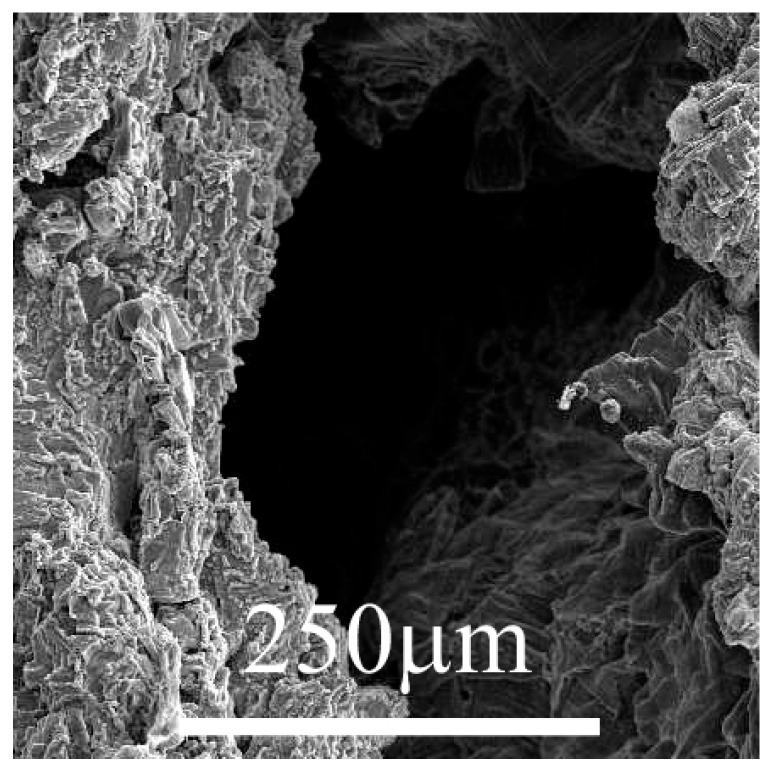
Poly (lactic-co-glycolic acid) (PLGA) porous scaffold for tissue engineering. Enlargement of a pore. The average pore size is 350–550 mm, and the porosity is estimated at 35–45%. © 2004–2024 University of Cambridge https://creativecommons.org/licenses/by-nc-sa/4.0/ (accessed on 15 December 2024).

## Data Availability

This study did not create or analyse new data, and data sharing is not applicable to this article.

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
