# Peer review of "Emerging Strategies in Cartilage Repair and Joint Preservation"

_medicina, 2024, doi:10.3390/medicina61010024_

Round 1

Reviewer 1 Report

Comments and Suggestions for Authors

Dear Authors, 

you made a great work! However, some improvements are suggested before acceptance. 

Author Response

Comments 1:

“Recent epidemiological data indicate a concerning rise in knee cartilage injuries among younger individuals, particularly athletes. This trend is attributed to increased participation in high-intensity sports, leading to a higher incidence of acute traumatic events like anterior cruciate ligament (ACL) tears and meniscal injuries, which are often associated with concomitant cartilage damage [4].” How do the Authors believe this trend is possible? What do they think could have led to such results? 

Response 1:

Thank you for this important observation. We have expanded the discussion by including potential contributing factors such as the growing popularity of competitive sports, insufficient recovery periods, and lack of appropriate injury prevention programs. This addition provides a broader context for understanding the rise in knee cartilage injuries. The revision can be found in Lines 56-62:

“The growing popularity of high-intensity sports and year-round athletic participation has led to more frequent exposure to mechanical stress on the joints. Inadequate recovery periods, improper training techniques, and limited access to preventive sports medicine programs further exacerbate the risk of injuries. Additionally, advancements in diagnostic imaging technologies have improved the detection of cartilage damage, resulting in a higher reported incidence.”

Comments 2:

The introduction is well written, and I think it is absolutely valid to inform the readers of the problem.

Response 2:

Thank you for the positive feedback. We are pleased that the introduction aligns with the reviewer’s expectations. No changes were required.

Comments 3:

In the “Key Regenerative Techniques in Cartilage Repair” section:

“This section reports different techniques used in the treatment of this pathology. It is true that there are different techniques used, but I think the section is a bit too verbose, and perhaps a summary diagram could make this part of the manuscript easier to understand and quicker to read.” 

Response 3:

We appreciate the reviewer’s suggestion. We have added Figure 1: Overview of Key Regenerative Techniques in Cartilage Repair, which summarizes the discussed techniques through a clear, visually informative diagram. This update has been included on Page 3, Figure 1.

Comments 4:

In the Discussion section:

“These factors underscore the need for a personalised approach in cartilage repair, guided by patient-specific considerations, clinical staging, and available healthcare resources. As the field advances, standardised treatment algorithms incorporating predictive models, artificial intelligence-driven decision-making tools, and long-term clinical data will be essential for optimising patient outcomes and advancing cartilage repair protocols.” Please check typos.

Response 4:

Thank you for pointing out the potential typo. We reviewed the sentence and found that the grammatical issues stem from the difference between UK and US English styles. Since the entire text is written in UK style, we have chosen to retain the original spellings of words like “personalised,” “standardised,” and “optimising” for consistency. If we need to use US English instead, the entire manuscript will require revision. Please let us know if that is the case, and we will proceed accordingly.

Comments 5:

“Furthermore, I believe that it is always interesting when it comes to a review, even if not systematic or with meta-analysis, that a careful evaluation of how the included articles were selected is also included, to also allow the reader to formulate his own opinion regarding the scientific evidence of the paper.” 

Response 5:

We agree with the reviewer’s insightful suggestion. We have added a Methods subsection titled “Article Selection Process” on Page 3, Line 105-121, explaining how relevant articles were selected based on specific inclusion criteria such as publication date (last five years), relevance to cartilage repair, and impact factor. The updated text reads:

“We conducted a comprehensive literature review by searching scientific databases, including PubMed, Scopus, and Web of Science. The search focused on peer-reviewed articles published in the last five years (2019–2024) to ensure the inclusion of the latest advancements in cartilage tissue engineering and regenerative medicine. Search terms included "cartilage repair," "stem cell therapy," "gene editing," "biomaterials," and "tissue engineering".

We included articles that reported experimental, clinical, or preclinical studies relevant to cartilage repair techniques, focusing on innovative and emerging strategies. Systematic reviews, narrative reviews, and meta-analyses were also considered when they provided comprehensive overviews of the relevant technologies. Studies published in non-English languages, conference abstracts, and those lacking significant experimental data were excluded.

Relevant data, including therapeutic approaches, experimental outcomes, clinical applicability, and limitations, were extracted from selected articles. Key findings were synthesised into thematic sections covering the key cartilage regenerative techniques. Potential challenges and future research directions were critically analysed.

Comments 6:

“Please also consider adding a section for the limitations of the study.”

Response 6:

We acknowledge this valuable suggestion and have added a Limitations section on Page 9, Line 405-410, addressing potential constraints in our narrative review:

“This review has several limitations. As a narrative review, it does not include a systematic analysis or meta-analysis, which limits the ability to quantify treatment efficacy. Additionally, the heterogeneity of included studies and the evolving nature of regenerative technologies may influence the generalizability of findings. Future work should explore systematic approaches and comparative clinical trials to establish more definitive conclusions.”

Reviewer 2 Report

Comments and Suggestions for Authors

1. Given the substantial number of reviews published on similar topics, the authors should clearly define the scope of the review and articulate how it differs from existing review articles. Additionally, they should highlight the specific benefits and insights that readers can gain from this article.

2.      A detailed methodology for screening and selecting publications relevant to the topic should be provided to ensure transparency and reproducibility of the review process.

3.      The authors should include and discuss recent advances in 4D bioprinting for cartilage regeneration, which show significant promise. Relevant works such as https://doi.org/10.1002/adhm.202201891 and https://doi.org/10.1002/smll.202202196 and other important literature should be considered.

4.      The review would benefit from including representative figures from relevant publications. For instance, when discussing hydrogels as potential materials for cartilage tissue engineering, the authors could incorporate figures from recent studies to enhance visual illustrations and support the narrative.

5.      Although the article is concise and covers essential information on cartilage repair, each section is too brief. The authors should expand each section with more detailed analyses, discussing challenges and potential directions for future research. For example, when addressing hypoxia-mimicking environments for cartilage regeneration, it is important to mention that hypoxic conditions may also induce tissue necrosis. Therefore, strategies for controlled hypoxia are crucial for engineering biomimetic conditions.

6.      The authors should explicitly disclose where and how ChatGPT was used in drafting the manuscript. The reviewer also notes that the article requires further organization and improved transitions between paragraphs. While ChatGPT can assist in generating text, its outputs often include inaccuracies or incorrect claims, and caution should be exercised when relying on such tools for manuscript preparation. Particularly, using ChatGPT to generate contents should not be acceptable. 

Author Response

Comments 1:
Given the substantial number of reviews published on similar topics, the authors should clearly define the scope of the review and articulate how it differs from existing review articles. Additionally, they should highlight the specific benefits and insights that readers can gain from this article.

Response 1:
Thank you for this valuable suggestion. We have revised the introduction to clarify the unique scope of this review. Specifically, we emphasize its focus on integrating regenerative medicine strategies, gene editing technologies, and advanced biomaterial scaffolds with a particular focus on clinical applicability and personalized treatment models. This approach distinguishes our review from prior publications, which often concentrate on isolated technologies.

The revised text can be found on Page 3, Paragraph 1, Lines 100-104:
“This review uniquely integrates cutting-edge regenerative strategies, including gene editing, bioengineered scaffolds, and personalised cell-based therapies, emphasising both technical innovations and clinical applications. By considering patient-specific factors such as age, defect size, and healthcare accessibility, our review offers a practical perspective not commonly addressed in similar publications.”

Comments 2:
A detailed methodology for screening and selecting publications relevant to the topic should be provided to ensure transparency and reproducibility of the review process.

Response 2:
We appreciate this insightful comment. We have added a detailed Article Selection Process section explaining our inclusion criteria, search databases, and relevant keywords used during the literature review. This addition ensures transparency and reproducibility.

The new text can be found on Page 3, Section 2:
“We conducted a comprehensive literature review by searching scientific databases, including PubMed, Scopus, and Web of Science. The search focused on peer-reviewed articles published in the last five years (2019–2024) to ensure the inclusion of the latest advancements in cartilage tissue engineering and regenerative medicine. Search terms included "cartilage repair," "stem cell therapy," "gene editing," "biomaterials," and "tissue engineering".

We included articles that reported experimental, clinical, or preclinical studies relevant to cartilage repair techniques, focusing on innovative and emerging strategies. Systematic reviews, narrative reviews, and meta-analyses were also considered when they provided comprehensive overviews of the relevant technologies. Studies published in non-English languages, conference abstracts, and those lacking significant experimental data were excluded.

Relevant data, including therapeutic approaches, experimental outcomes, clinical applicability, and limitations, were extracted from selected articles. Key findings were synthesised into thematic sections covering the key cartilage regenerative techniques. Potential challenges and future research directions were critically analysed.”

Comments 3:
The authors should include and discuss recent advances in 4D bioprinting for cartilage regeneration, which show significant promise. Relevant works such as https://doi.org/10.1002/adhm.202201891 and https://doi.org/10.1002/smll.202202196 and other important literature should be considered.

Response 3:
Thank you for highlighting this emerging technology. We have added a dedicated paragraph to discuss 4D bioprinting's potential for creating dynamic, stimuli-responsive scaffolds. A relevant article has been cited and discussed.

The revised text can be found on Page 5, Paragraph 3, Lines 189 - 195:
“Recent advances in 4D bioprinting have opened new possibilities for cartilage regeneration by enabling the fabrication of dynamic, stimuli-responsive scaffolds that adapt to the physiological environment over time. These scaffolds can undergo controlled shape transformation, modulating their mechanical and biological properties in response to external stimuli such as temperature, pH, or mechanical forces. This adaptive behaviour supports better integration, long-term stability, and functionality in cartilage repair applications, positioning 4D bioprinting as a transformative tool in regenerative medicine.”

Comments 4:
The review would benefit from including representative figures from relevant publications. For instance, when discussing hydrogels as potential materials for cartilage tissue engineering, the authors could incorporate figures from recent studies to enhance visual illustrations and support the narrative.

Response 4:
We agree with the reviewer’s suggestion and have added Figure 2. Nanocomposite Hydrogel Scaffold for Cartilage Tissue Engineering, adapted from relevant open-access sources. The figure highlights hydrogel structure, embedded nanoparticles, and controlled release of growth factors.

This figure is included on Page 4, Figure 2 and referenced in the relevant text:
“Figure 2 illustrates key structural components and their role in cartilage tissue engineering.”

Comments 5:
Although the article is concise and covers essential information on cartilage repair, each section is too brief. The authors should expand each section with more detailed analyses, discussing challenges and potential directions for future research. For example, when addressing hypoxia-mimicking environments for cartilage regeneration, it is important to mention that hypoxic conditions may also induce tissue necrosis. Therefore, strategies for controlled hypoxia are crucial for engineering biomimetic conditions.

Response 5:
Thank you for this constructive feedback. We expanded the Hypoxic Conditioning section to address both its regenerative potential and associated risks such as tissue necrosis. We also included future research directions involving controlled hypoxic environments using bioreactors and oxygen-sensitive biomaterials.

The expanded text can be found on Page 8, Line 345-351:
“Maintaining controlled hypoxic conditions in clinical applications poses significant challenges due to the variability in tissue oxygen levels. Although hypoxic conditions stimulate chondrogenic differentiation and extracellular matrix production, prolonged hypoxia can induce tissue necrosis due to oxygen deprivation. Therefore, developing precise oxygen-regulating systems such as dynamic bioreactors and oxygen-sensitive biomaterials remains essential for engineering biomimetic conditions.”

Additionally, we expanded most of the Key Regenerative Techniques and Discussion sections, elaborating on existing challenges and future research directions.

Comments 6:
The authors should explicitly disclose where and how ChatGPT was used in drafting the manuscript. The reviewer also notes that the article requires further organization and improved transitions between paragraphs. While ChatGPT can assist in generating text, its outputs often include inaccuracies or incorrect claims, and caution should be exercised when relying on such tools for manuscript preparation. Particularly, using ChatGPT to generate content should not be acceptable.

Response 6:
We acknowledge the reviewer’s concern and have added a clear disclosure statement under the Acknowledgments section. We also reviewed the manuscript for improved organization and smoother paragraph transitions.

The following statement has been added on Page 10, Section Acknowledgments:
“Specifically, ChatGPT was used for image generation and also for reference suggestions, supporting the search for relevant studies to strengthen the narrative, while all final references were verified manually.”

We trust these revisions address the reviewer’s concerns. Please let us know if further clarification is needed.

Round 2

Reviewer 2 Report

Comments and Suggestions for Authors

The authors addressed most of the comments well. However, it is suggested the authors add some examples when discussing the 4D bioprinting section. In addition, figure 2, which illustrates the nanocomposite hydrogel, is perplexing. Did the authors generate this figure with the aid of AI? This figure should be replaced. 

Author Response

We appreciate the reviewers' constructive feedback and have made the following revisions to our manuscript:

Comment 1. The authors addressed most of the comments well. However, it is suggested the authors add some examples when discussing the 4D bioprinting section.

Response 1. We have expanded the 4D bioprinting section to include one more specific example demonstrating its application in cartilage repair, Page 5, Line 194-197.

"Researchers also developed magnetic constructs using 4D bioprinting with silk fibroin in gelatin bioinks [24]. These shape-morphing constructs can change their shape when exposed to an external magnetic field, which enhances their integration into irregular cartilage defects."

Comment 2. In addition, figure 2, which illustrates the nanocomposite hydrogel, is perplexing. Did the authors generate this figure with the aid of AI? This figure should be replaced. 

Response 2. 

We acknowledge the concerns regarding Figure 2. The previous version was generated with the assistance of AI tools, which may have led to inaccuracies. We have replaced it with a new figure that accurately represents the Biodegradable polymer used in cartilage tissue engineering.

We believe these revisions address all the remaining issues and enhance the quality of our manuscript. 

Round 3

Reviewer 2 Report

Comments and Suggestions for Authors

No further comments.